# Drought and Plant Community Composition Affect the Metabolic and Genotypic Diversity of *Pseudomonas* Strains in Grassland Soils

**DOI:** 10.3390/microorganisms9081677

**Published:** 2021-08-07

**Authors:** Claudia Breitkreuz, Thomas Reitz, Elke Schulz, Mika Tapio Tarkka

**Affiliations:** 1Department of Soil Ecology, Helmholtz Centre for Environmental Research GmbH-UFZ, Theodor-Lieser-Str. 4, 06120 Halle, Germany; thomas.reitz@ufz.de (T.R.); elke.schulz2@gmx.net (E.S.); mika.tarkka@ufz.de (M.T.T.); 2German Centre for Integrative Biodiversity Research (iDiv) Halle-Jena-Leipzig, 04103 Leipzig, Germany

**Keywords:** *Pseudomonas*, grassland, phosphate solubilization, potassium solubilization, siderophore production

## Abstract

Climate and plant community composition (PCC) modulate the structure and function of microbial communities. In order to characterize how the functional traits of bacteria are affected, important plant growth-promoting rhizobacteria of grassland soil communities, pseudomonads, were isolated from a grassland experiment and phylogenetically and functionally characterized. The Miniplot experiment was implemented to examine the mechanisms underlying grassland ecosystem changes due to climate change, and it investigates the sole or combined impact of drought and PCC (plant species with their main distribution either in SW or NE Europe, and a mixture of these species). We observed that the proportion and phylogenetic composition of nutrient-releasing populations of the *Pseudomonas* community are affected by prolonged drought periods, and to a minor extent by changes in plant community composition, and that these changes underlie seasonality effects. Our data also partly showed concordance between the metabolic activities and 16S phylogeny. The drought-induced shifts in functional *Pseudomonas* community traits, phosphate and potassium solubilization and siderophore production did not follow a unique pattern. Whereas decreased soil moisture induced a highly active phosphate-solubilizing community, the siderophore-producing community showed the opposite response. In spite of this, no effect on potassium solubilization was detected. These results suggest that the *Pseudomonas* community quickly responds to drought in terms of structure and function, the direction of the functional response is trait-specific, and the extent of the response is affected by plant community composition.

## 1. Introduction

Species of the genus *Pseudomonas* are ubiquitously present in soils. More than 220 species have been reported in the literature from this diverse genus, with validated taxonomic names [1]. Capable of utilizing a wide range of organic and inorganic compounds, expressing both genetic and metabolic versatility, and being relatively easy to isolate and cultivate, they are among the best-studied bacteria in soil [2]. Several root-associated *Pseudomonas* strains are able to stimulate plant growth, and therefore are classified as plant growth-promoting rhizobacteria (PGPR). The beneficial effects rest upon various direct and indirect mechanisms, such as inducing or increasing plant disease resistance [3], producing phytohormones such as auxin [4], or decreasing plant ethylene levels and, thus, plant stress by ACC-deaminase production [5]. In addition to this, pseudomonads play a crucial role for plant nutrient acquisition [6]. In fact, the majority of essential micro- and macronutrients in soils are bound in mineral and organic complexes, and in this form are not available for plants. For instance, in an average soil, P content is about 0.05% (*w*/*w*), from which only 0.1% is in an available form for direct uptake [7]. The contribution of pseudomonads to mineral P and also K solubilization is primarily based on the exudation of organic acids [8,9,10], which lower the pH and thus accelerate the process of weathering [10,11,12]. Unlike the need for the macronutrients P and K, plant demand for iron is lower. Nevertheless, iron is often poorly available in soils. Some species of the genus *Pseudomonas* can mobilize iron by exuding siderophores, which are low-molecular weight compounds with a high affinity for Fe(III)-ions [13], thus supporting iron nutrition of plants [14,15]. Notwithstanding their relevance for ecosystem productivity and functioning, the impact of the ongoing climate change on *Pseudomonas* communities in soils is still not fully resolved.

Warming and changed precipitation have a profound impact on the structural and functional diversity of soil microbial communities [16,17], i.e., not only on the structure of soil microbial communities, but also on microbe-mediated soil processes, such as decomposition or nutrient mineralization. Besides the direct effects caused by changed temperatures and soil moisture, soil microorganisms are also affected by indirect impacts of climate change, mainly in the form of altered plant community composition [18,19]. Due to the changed climatic conditions, the geographical distribution patterns of plant populations change and plant species establish themselves in novel regions to maintain optimal conditions for growth [20]. For instance, the plant community composition of endemic grassland floras in Central Europe are expected to change to resemble communities currently established in Southern Europe. Shifts in plant community composition lead to alterations in the quality and quantity of root exudates as well as plant litter composition. Under these unsettled conditions, the abundance of microbial generalists in the soil might be promoted, which can cope with a broad range of resources, and the abundance of specialists might be reduced [21]. For pseudomonads, the *P. putida* and *P. koreensis* strains, which have previously been reported as common members of the plant growth-promoting and drought-tolerant *Pseudomonas* community [22,23], are expected to dominate the *Pseudomonas* community. In this respect, the mediating role of plant community composition on the impact of climate change on soil pseudomonades, such as important plant growth-promoting soil bacteria [24,25], is of particular interest.

To study the structural and functional responses of soil pseudomonads to climate change, we used the “Miniplot” grassland experiment located in Bad Lauchstädt (Germany), which was designed to examine the mechanisms underlying grassland ecosystem changes due to climate change. Two factors directly and indirectly related to climate change were manipulated, namely precipitation pattern (drought vs. ambient rain) and plant community composition (mixtures of grassland species with a predominant distribution in Northeast (NE) or Southwest (SW) Europe). Thereby, the NE species are seen as potential losers due to climate change in the experiment, expected to have a lower adaptability to longer drought periods and higher temperatures. In contrast, the SW plant community may buffer the impact of drought stress on the microbial community, since they are adapted to warmer and dryer conditions and therefore can be seen as the potential winners in the experiment.

We hypothesized that the *Pseudomonas* community will be dominated by generalists, which can cope with the conditions of different plant community compositions and drought. In the second hypotheses, we stated that severe drought will influence the *Pseudomonas* community composition and reduce the activity of P, K and Fe-mobilizing *Pseudomonas* strains. Finally, we hypothesized that, due to their better adaptation to drought, the SW plant communities mitigate the negative impact of drought on the *Pseudomonas* community. To address these hypotheses, a total of 464 *Pseudomonas* strains were isolated from the Miniplot experiment, grouped into phylogenetic clusters and tested for their potential to solubilize tricalcium phosphate and potassium silicate, as well as to produce siderophores.

## 2. Materials and Methods

### 2.1. Experimental Design and Soil Sampling

The so called “Miniplot experiment” was established in 2010 at the field research station of the Helmholtz-Center for Environmental Research in Bad Lauchstädt (central Germany, 11°53′E, 51°24′N). The site is characterized by an annual precipitation of about 480 mm and a mean annual temperature of 8.7 °C. The experiment comprised a total of 80 plots (1 m^2^), delimited to each other by stainless steel frames up to a depth of 60 cm (Appendix A). The plots were filled with soil from the experimental site, which is characterized as a humus- and nutrient-rich Haplic Chernozem [26]. To induce drought, half of the plots were roofed in 2012 for five weeks in May/June as well as for six weeks in July/August/September (D, drought treatment), whereas the other half of the plots were exposed to ambient precipitation (C, control treatment). The total amount of intercepted precipitation was 96.2 and 70.8 mm during the first and second roofing periods, respectively. The plant treatment comprised two pools of either 32 Northeast European (NE) or Southwest European (SW) plant species (eight grasses, eight small herbs, eight tall herbs and eight legumes). The 80 plots were sown with either 16 randomly partitioned Northeast European (NE) or Southwest European (SW) species (four of each functional group), or with a representative mix (8 NE, 8 SW) of both plant species pools (Appendix A). Soil samples were collected from 15 control and 15 drought plots (5 SW, 5 NE, 5 Mix, each), which showed the lowest weed fraction in the plant communities (Appendix A).

Soil samples were taken immediately after the roofing periods, i.e., on the 27th of June and the 6th of September in 2012. For this, 18 soil cores (0–10 cm, 12 mm diameter) from each plot were pooled, sieved (2 mm) and stored at −20 °C for soil chemical analysis, as well as for the isolation of pseudomonads.

### 2.2. Soil Parameters

Soil carbon (C) and nitrogen (N) contents were determined by dry combustion in triplicate using a Vario EL III C/H/N analyser (Elementar, Hanau, Germany). Since the carbonate concentration of the soils was negligible (<2%), the total C concentration measured was considered to represent total organic carbon (TOC). Available soil phosphorus was determined after extraction with double lactate (pH 3.6, 90 min; [27]) using the molybdenium blue method [28]. Gravimetric soil moisture contents were determined using a fully automatic moisture analyzer (Kern DBS60–3, Kern & Sohn GmbH, Germany). Soil pH was measured with a pH electrode (Mettler Toledo InLab Expert Pro-ISM) after shaking the soil for 1 h in 0.01 M CaCl_2_ (1:2.5 *w*/*v*).

At the time of soil sampling in June 2012, the average gravimetric soil moisture was 7.7% in the drought and 15.1% in the control samples and, in September, 6.7% in the drought and 11.1% in the control samples (Appendix A). Soil properties were comparable between seasons, plant community compositions and drought or control treatments. In addition, TOC increased from June to September, and P availability was higher in plots with drought compared to control treatment (Appendix A).

### 2.3. Selective Isolation and Cultivation of Pseudomonas Bacteria

For selective isolation of *Pseudomonas* spp. from the soil samples, we used “*Pseudomonas* Selective Isolation Agar (PSIA)” [29]. It was described as being highly specific for pseudomonads suppressing other bacterial and fungal growth by the addition of crystal violet (2 mg/L), nitrofurantoin (350 mg/L) and cycloheximide (100 mg/L) to soybean casein digest agar. From each sample, 0.5 g of soil was suspended in 50 mL of sterile water, stirred with a magnetic stirrer for 15 min, and incubated at room temperature for 20 min. Three replicates of 100 µL supernatant were plated on PSIA. Inoculated agar plates were incubated at 25 °C for three days. Up to 15 colonies were picked per soil sample. The isolated strains were cultivated on YME agar (4 g/L yeast extract, 10 g/L malt extract, 4 g/L glucose and 20 g/L agar).

### 2.4. DNA Extraction and Partial 16S rRNA Gene Sequencing

Three-day-old colonies from YME agar were transferred into Eppendorf tubes with 300 µL of 60% polyethylenglycol 200 and a glass bead. The bacterial cell walls were mechanically lysed by vortexing and the cell suspensions were 1:50 diluted with sterile water before PCR amplification. 16S rRNA gene fragments were amplified using Promega Green (Promega, Madison, WI, USA) with primers 27f (10 µM–5′-AGAGTTTGATCMTGGCTCAG-3′ [30]) and 1492r (10 µM—5′-GGTTACCTTGTTACGACTT-3′ [30]) with the following PCR program: initial denaturation at 95 °C for 7 min, 4 cycles of denaturation at 94 °C for 30 s, annealing at 56 °C for 30 s, elongation at 72 °C for 75 s, 31 cycles of denaturation at 94 °C for 30 s, annealing at 54 °C for 30 s, elongation at 72 °C for 75 s, and a final elongation step at 72 °C for 15 min. After quality checking PCR products on a 1.5% agarose gel, they were purified with ExoSAP (Affymetrix, Santa Clara, CA, USA) and cycle sequenced with BAC 341f primer (10 µM—5′-CCTACGGGAGGCAGCAG-3′ [31]) using Big Dye Termination Mix (GeneCust Europe, Dudelange, Luxemburg). DNA was ethanol precipitated, dried and solubilized in highly deionized formamide. DNA fragments were visualized by an ABI 3730xl DNA Analyzer (Applied Biosystems, Foster City, CA, USA) and quality checked by manually comparing the sequences with the individual chromatograms in BioEdit (Ibis Biosciences, Carlsbad, CA, USA). All 16S rRNAgene sequences were deposited in the NCBI database: MZ541103-MZ541566.

For cluster analysis, sequences were trimmed in BioEdit to a defined length of 720 nucleotides starting from the sequence position 387 of E. coli 16S rRNA gene. Trimmed sequences were clustered with CD-Hit Est [32] using an identity cut off 99.5%. Phylogenetic trees of cluster and reference sequences were constructed with Neighbor Joining Method (NJ, bootstrap = 1000) in MAFFT server [33] and drawn with Archaeopterix [34].

### 2.5. Quantification of P- and K- Solubilizing Activity and Siderophore Production

Phosphorus solubilization activities of the bacterial isolates were tested with three replicates for each strain on Pikovskaya medium [35]. This consisted of 10 g/L glucose, 0.2 g/L NaCl, 0.5 g/L (NH_4_)_2_SO_4_, 0.2 g/L KCl, 2 mg/L FeSO_4_ × 7 H_2_O, 2 mg/L MnSO_4_ × H_2_O, 5 g/L Ca_3_(PO)_4_, 10 mg/L bromophenol blue and 15 g/L agar, pH 7. The insoluble tri-calcium phosphate caused the medium to have a milky appearance. The solubilizing activities were assessed by clear zones around the bacterial colonies, termed “halo” in the following text.

Siderophore production was investigated with three replicates on CAS agar according to Louden et al. [36]. In the medium, the ferric iron is tightly bound to CAS/HDTMA, producing the green-blue color of the medium. The color of the medium changes to orange when the iron–CAS/HDTMA complex is dissolved by the siderophores. After three days (iron) or two weeks (phosphorus) of incubation at 25 °C, the diameter of the halos around the colonies and the diameter of the bacterial colonies were measured. Activity levels were defined by calculating the area of the halo formation (including the colonies), as this area is correlated with the amount of released P and the amount of released iron from the CAS/HDTMA complexes by siderophores.

Potassium solubilization from K-feldspar was investigated with three replicates in liquid Aleksandrov medium [37], which consisted of 5 g/L glucose, 0.5 g/L MgSO_4_ × H_2_O, 0.1 g/L CaCO_3_, 6 mg/L FeCl_3_, 2 g/L NaH_2_PO_4_, and 3 g/L K-feldspar. Bacterial colonies were suspended with an inoculation loop in 400 µL sterile water and subsequently 100 µL of the bacterial suspension were used to inoculate 15 mL Aleksandrov medium in 50 mL conical tubes. The tubes were placed on a shaker (100 rpm) and incubated at 25 °C for two weeks with intermittent opening under sterile conditions to achieve gas exchange after one week. After incubation, the tubes were centrifuged at 9000 rpm to remove K-feldspar and cells from the medium and the content of soluble potassium in the supernatant was determined with a K+ sensitive electrode (perfectION, Mettler-Toledo, Gießen, Germany).

### 2.6. Statistical Analysis

All statistical analyses were performed in R version 3.6.2 [38]. The effects of drought treatment, plant community composition as well as the sampling date (season) on the relative abundance and activities of *Pseudomonas* strains were assessed using analysis of variance (ANOVA). Tukey’s honestly significance difference (HSD) post hoc test was then implemented to find means that are significantly different from each other. The effects of soil parameters and biomass on the activities were assessed using Spearman’s rank correlations.

## 3. Results

### 3.1. Isolation of Pseudomonas Strains

Using a *Pseudomonas*-selective isolation agar (PSIA), in total, 604 bacterial isolates were obtained from soil samples taken during the June and September sampling. Out of the 604 isolates, 464 (77%) were assigned to the genus *Pseudomonas* by partial sequencing of the 16S rRNA gene, which demonstrated the high selectivity of the PSIA medium for pseudomonads (full list of isolates provided in Appendix A). We isolated 247 and 217 *Pseudomonas* strains from the June and September sampling, respectively. Throughout the two sampling points, 265 and 199 strains were obtained from soil samples of control and drought plots, as well as 152, 149 and 163 strains from plots with NE, Mix and SW plant species, respectively.

### 3.2. Phylogenetic Classification and Distribution of the Isolates

Partial 16S rDNA gene sequencing, sequence clustering and phylogenetic tree construction were used to reveal the phylogenetic relationships among the *Pseudomonas* isolates. To avoid biases due to low-quality sequences, only sequences with at least 720 bp were considered (excluding five sequences, *n* = 459). Cluster analysis with a similarity threshold of 99.5% assigned the *Pseudomonas* 16S sequences to 14 clusters. A phylogenetic tree was constructed by neighbor-joining method with one representative sequence from each cluster as well as corresponding reference sequences (Figure 1). The analysis revealed that the vast majority of isolated strains were related to *P. koreensis* (cluster C1), *P. putida* (C2, more distant C4), *P. helmanticensis* (C3) and *P. vancouverensis* (C5). Rare sequence types represented by only one strain each were also identified, such as *P. chlororaphis* (C11), *P. abietaniphila* (C13) and *P. azotoformans* (C14).

The relative abundance of the *Pseudomonas* clusters was influenced by drought treatment and season (Figure 2). For instance, strains associated with clusters C1 and C5 were isolated more often from plots with drought treatment, whereas C2 and C4 representatives were primarily obtained from ambient precipitation plots. While the C4 strains were primarily isolated in June, most of the C2 strains were obtained from the September sampling. Plant community composition also had an effect on cluster representation. Fewer C1 sequences were isolated from plots with SW than from plots with Mix or NE plant communities, and for C2 sequences, the relative abundance was decreased in Mix plant community plots.

### 3.3. Impact on Functionality of the Isolated Pseudomonas Strains

P-solubilization activity (Figure 3) of strains from the experimental plots with drought treatment was higher (*p* = 0.01) than that of strains isolated from control plots, whereas an opposing pattern was observed for siderophore production levels (*p* < 0.001). This was related to soil moisture contents and available phosphate concentrations (Table 1).

The negative impact of drought for siderophore production was prominent in both June and September. Regarding the influence of plant community composition, K- (*p* = 0.02) and *P*-solubilizing (*p* = 0.008) activities were highest in NE plant community plots. Season influenced the pseudomonads’ activity only to a minor extent, but the results reveal a higher K-solubilization activity in September than in June (*p* = 0.01). Furthermore, interactions between plant community composition and drought treatment indicated lower K solubilization activity under drought conditions in mixed communities, but no differences in the response between SW and NE plant communities were observed (*p* = 0.03).

### 3.4. Relationships between Phylogenetic Classification and Functional Traits of the Isolates

The determined activity traits of the isolated *Pseudomonas* strains were related to their respective 16S rRNA cluster representation for the most abundant clusters, C1–C5 (Figure 4). The results reveal a relationship between functional and structural diversity. For P-solubilization, the highest activities were found for clusters 1, 3 and 5, whereas the activities in cluster 2 and 4 were significantly lower. Siderophore production was high for strains in clusters 2 and 5 and low for 1 and 4. Average K-solubilization activity was almost completely unaffected by structural diversity (Figure 4).

## 4. Discussion

While soil moisture and plant species distribution affect the structure and function of bacterial communities, the respective and interacting impacts on specific genera with great importance for plant nutrient acquisition have not been determined. The present study shows that the abundance, phylogenetic composition and traits of the soil *Pseudomonas* community are affected by drought periods and differences in plant community composition, and that these changes are affected by season.

### 4.1. Phylogenetic Identity of Pseudomonads

Pseudomonads were grouped into 14 clusters, and each represented distinct species of the genus. According to our first hypothesis stating that frequently isolated generalists occur in the *Pseudomonas* community, clusters 1–5 represented species with occurrence in all or most treatments, and clusters 6–14 represented less abundant taxa. The functional potential of these rare strains in culture collection should not be underestimated. For instance, the two strains of cluster 11 had the highest identity cover with *Pseudomonas chlororaphis* 16S rDNA and were only isolated from one plot of the experiment. Bloemberg and Lugtenberg [39] categorized strains of this species as efficient root colonizers and producers of antifungal substances such as phenazine-1-carboxamide (PCN), cyanide, chitinases and proteases, indicating that cluster 11 strains should be tested against plant pathogens.

The most abundant clusters (cluster 1–5) included generalists such as *P. koreensis* (166 isolates) and *P. putida* (85 strains), which both exert plant growth-promoting activities. In a recent study, the interacting effects of potassium-solubilizing *P. koreensis* and phosphate-solubilizing *P. putida*, among others, as a bacterial fertilizer could improve the performance of basil under water limiting conditions [40]. *P. koreensis* has been characterized from agricultural soil [41]. Many strains of this species express plant growth-promoting activities, such as P and K solubilization, as well as siderophore and IAA production [4,42]. *P. putida* is a widely studied species with a highly versatile metabolism, that solubilizes K and P [43,44], and produces siderophores [45]. Mazzola and Gu [46] reported that wheat cropping of apple orchard soils led to an increase in the proportion of *P. putida strains*, but in our work, we did not find any PCC-dependent distribution of the *P. putida.*

### 4.2. Effects of the Experimental Treatments on Functional Properties of the Pseudomonas Community

Some studies have indicated that the soil microbial populations in a grassland ecosystem are resilient to climatic extremes [47,48,49]. This suggests the presence of microorganisms adapted to regular, seasonal fluctuations in temperature and precipitation. According to this suggestion, we observed an effect of season on K-solubilizing activities, as well as a moisture-related activity shift in the *Pseudomonas* community resulting in a more active phosphate-solubilizing population. Drought decreases P availability to plants and microorganisms [50], suggesting that the increased potential for P solubilization by the pseudomonads may be important for P availability. The suggestion that biological P mineralization is beneficial for plant growth is supported by the work of Marasco et al. [25], who presented evidence that the recruitment of P-solubilizing bacteria by root systems is an important factor for plant survival in a desert ecosystem. In contrast, the siderophore production potential of the strains decreased during drought treatments in all PCC types. Iron acquisition is necessary for bacterial growth [51], and since the synthesis of siderophores is energy-intensive [52], it is under the control of available iron. When the availability of iron is low in dry soils, one would expect an increase in siderophore production potential by the bacteria from roofed plots. As such, siderophore-producing *Azotobacter* strains increased iron concentrations in maize leaves after inoculation and when exerted to drought conditions [53]. Co-inoculation of *Bacillus amyloliquefaciens NBRISN13* and *Pseudomonas putida NBRIRA* with multiple beneficial traits, e.g., mineral solubilization and siderophore production, ameliorated drought stress response in chickpea [54], whereby specific mechanisms could not be fully resolved. In contrast, our data indicate that the *Pseudomonas* community may be iron limited in dry soil due to reduced siderophore production.

In this dataset, modified plant community composition was clearly a less impacting factor than soil moisture structuring the *Pseudomonas* population. In accordance with our observations, Latour et al. [55] revealed that both the host plant and the soil type affected the *Pseudomonas* community profile. Contrary to that, Schreiter et al. [56] investigated specific traits of *Pseudomonas sp. RU47* in a pot experiment, and found that functionality was not affected by either plant species or soil type at all. However, our third hypothesis, stating that the negative impacts of drought on *Pseudomonas* community might be mitigated in SW plant community composition plots, could not be confirmed, as we did not find an interacting effect of drought and SW. Furthermore, the K- and P-solubilizing activity was strongest in strains from the endemic NE plant communities. This result may be based on either the different litter or rhizodeposit quality of NE and SW plant communities [57,58], and suggests better adaptation of the K- and P-solubilizing bacteria to endemic NE plant communities than SW communities.

### 4.3. Links between Phylogeny and Function

Partial agreement between 16S rRNA cluster distribution and metabolic properties of the corresponding strains suggests that the ability to solubilize P or to produce siderophores for iron acquisition can, in part, be predicted from *Pseudomonas* phylogeny. In general, such correspondence has been suggested for such genetically simple traits and taxonomic position of bacteria [59]. When Latour et al. [55] investigated plant root-associated *Pseudomonas* populations, the phenotypic clustering of isolates, based on the growth on different carbon sources, correlated well with genotype patterns. By contrast, no clear relationship between the distribution of the metabolic types and the distribution of *Pseudomonas* species was found by Clays-Josserand et al. [60]. The 16S phylogeny and the metabolic property datasets were also not in agreement in a collection of pseudomonads from alpine soils, an environment that is characterized by dramatic seasonal shifts in physical and biochemical properties, and a heterogeneous resource distribution [61]. We assume that the level of diversity–function relationships of pseudomonads depends on the function in question, and may be influenced by environmental parameters.

In this report and in accordance with our assumptions in hypothesis two, the relative abundances of the more closely related clusters 1 and 5 decreased, as well as those of 2 and 4 increased under drought conditions compared to control plots. Interestingly, these combinations showed a complementary behavior at the two different sampling time points. As such, the fact that each cluster was either dominant during June (cluster 4 and 5) or September sampling (cluster 1 and 2) suggests seasonal variation among the *Pseudomonas* taxa. Seasonal differences may be associated with changes in the quantity and quality of organic carbon available from plant litter inputs or root exudation [62,63], but also with responses to temperature- and precipitation-induced changes in soil edaphic and plant phenotypic properties [64]. Recently, we investigated the potential of cultivable bacterial species for phosphate solubilization in the rhizosphere of winter wheat. Wheat stem elongation stage was associated with a high abundance of *Pseudomonas*, but also at the grain filling stage with *Phyllobacterium* [65]. Since the members of the clusters 1 and 2 also responded to PCC, both plant-derived compounds and changes in soil properties and temperature may play out in the differential seasonal distribution of this group.

## 5. Conclusions

We linked the structural and functional response of the soil *Pseudomonas* community to two drivers of global change: prolonged drought periods in the growing season and climate change-affected plant community structure. Our work demonstrated the negative impact of drought on the number and activity of siderophore producers, but increased activity for inorganic phosphate solubilization, which is an important trait for plant growth promotion. Contrary to our assumptions, SW plant communities did not mitigate effects of drought on bacterial activity levels. This might indicate a strong adaptation of the *Pseudomonas* community to the endemic NE plant communities to promote plant growth under drought conditions. However, in a next step, individual strains or consortia should be tested in a pot experiment or under field conditions to prove the observed patterns and possible plant growth promotion.

## Figures and Tables

**Figure 1 microorganisms-09-01677-f001:**
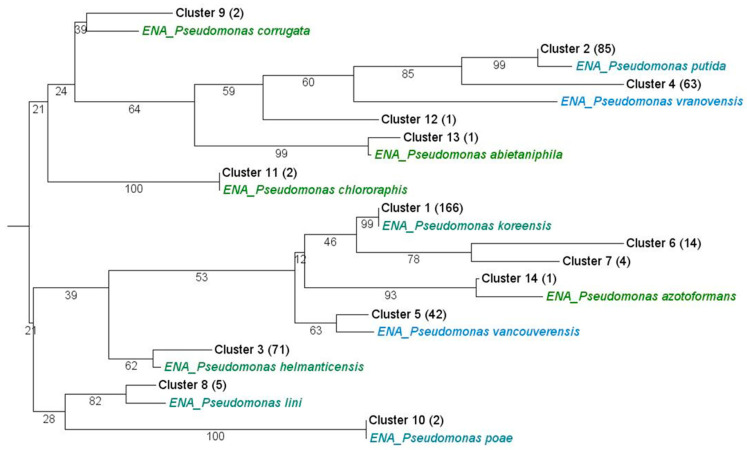
Phylogenetic classification. Unrooted neighbor-joining tree based on nucleotide differences between partial 16S rRNA gene sequences of the isolates as well as closely related strains. All sequences were separated into 14 clusters using an identity threshold of 99.5%. Numbers after the cluster designation indicate the number of sequences in each cluster, and numbers at the branches the bootstrapping values.

**Figure 2 microorganisms-09-01677-f002:**
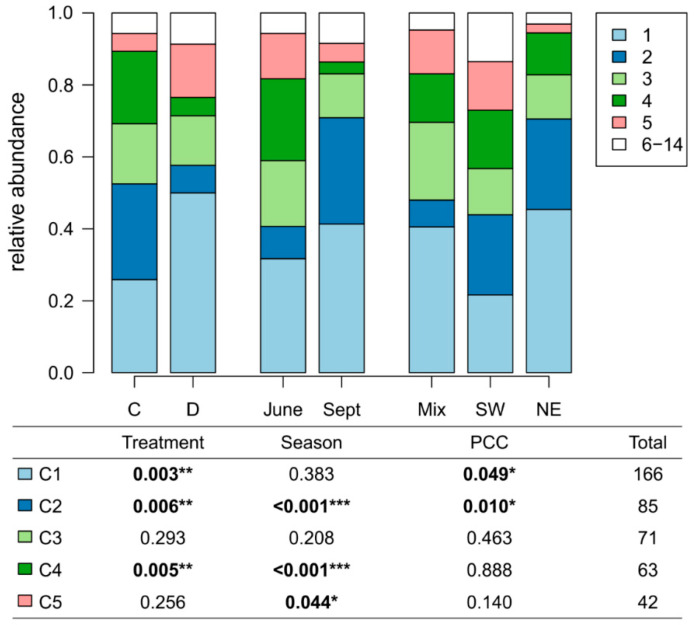
Drought, plant community composition and season influence the 16S rRNA diversity of *Pseudomonas* community. Abundances of strains representing fourteen 16S rRNA gene clusters are shown. The clusters 6–14 are grouped due to their low abundance. Five most abundant 16S rRNA gene clusters were closely related to the genes of *P. koreensis* (cluster C1), P. putida (C2), *P. helmanticensis* (C3), *P. putida* (C4, more distant C1) and *P. vancouverensis* (C5). Significant differences according to the ANOVA test are reported using the following convention: *** *p*  <  0.001, ** *p* < 0.01, * *p*  <  0.05.

**Figure 3 microorganisms-09-01677-f003:**
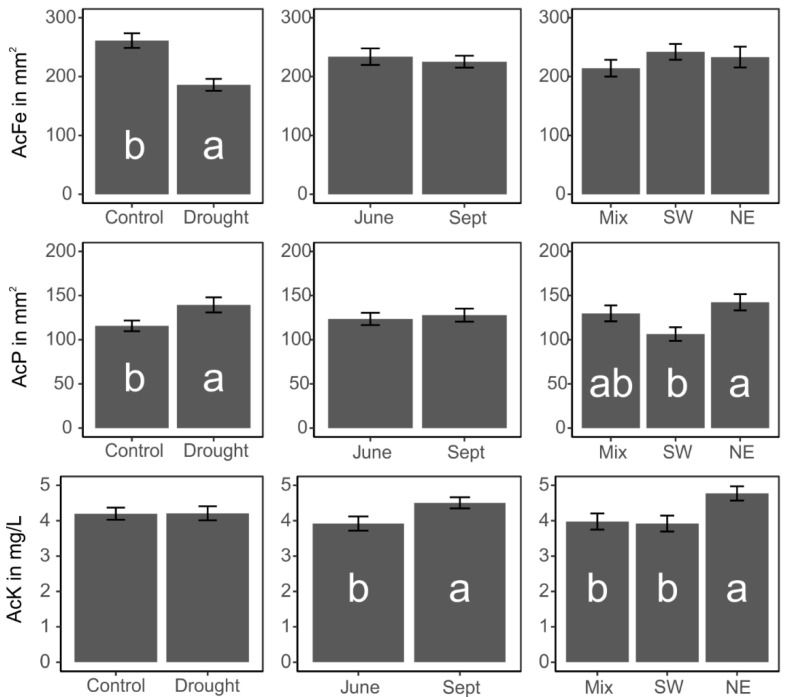
Mean ± standard error biological activities of the isolated *Pseudomonas* strains. Influence of treatment (C, control; D, drought), plant community composition (SW, Southwest European plant species; NE, Northeast European plant species; Mix, SW and NE plant species) and season (June and Sept, September) on potassium (AcK) and phosphate (AcP) solubilization, as well as siderophore (AcFe) production activities of the *Pseudomonas* strains. Significant differences (*p* <0.05) according to ANOVA and Tukey’s HSD test are indicated by different letters.

**Figure 4 microorganisms-09-01677-f004:**
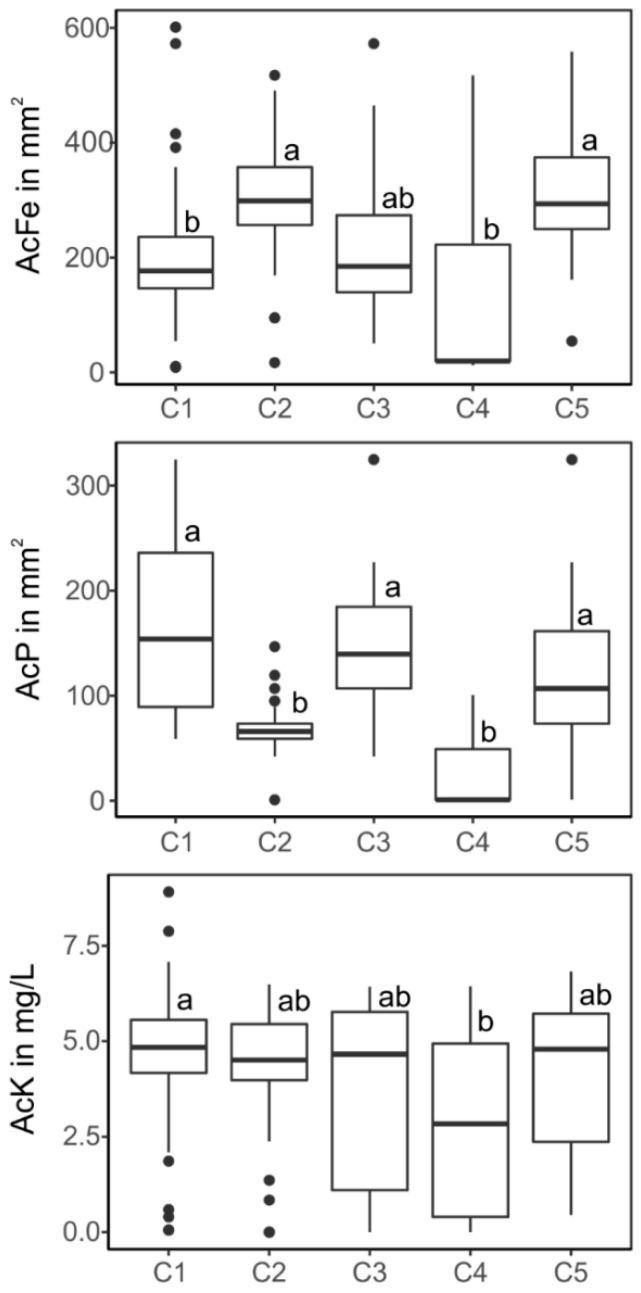
Activities for phosphate and potassium solubilization, as well as siderophore production from *Pseudomonas* strains among the most abundant clusters 1 to 5. Different letters indicate significant (*p* < 0.05) differences between clusters according to Tukeys HSD test.

**Table 1 microorganisms-09-01677-t001:** Impact of soil parameters on the biological activities of *Pseudomonas* bacteria. Correlation coefficients and significance of correlation are given. The significance of the positive or negative Spearman’s rank correlation coefficient is reported using the following convention: * *p*  <  0.05. The following datasets were used for analysis: AcK, AcP and AcFe, K and P solubilization and siderophore production activities of *Pseudomonas* strains; gravimetric soil moisture; pH; Soil-P, plant available phosphate; TOC and TN, total organic carbon and nitrogen, as well as plant biomass (PB) and root biomass (RB).

	pH		Moisture		Soil P		TOC		TN		PB		RB	
**AcFe**	**−0.22**	**0.10**	**0.34**	**0.01 ***	**−0.28**	**0.04 ***	−0.11	0.41	−0.11	0.44	0.18	0.18	0.02	0.88
**AcP**	0.10	0.46	**−0.33**	**0.01 ***	0.23	0.09	0.05	0.70	0.11	0.41	−0.18	0.20	−0.03	0.83
**AcK**	−0.16	0.23	−0.09	0.52	0.15	0.29	0.08	0.58	0.11	0.42	0.21	0.13	0.14	0.32

## Data Availability

16S rRNA gene sequences were deposited in the NCBI database: MZ541103–MZ541566.

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
