# Peer review of "Drought and Plant Community Composition Affect the Metabolic and Genotypic Diversity of Pseudomonas Strains in Grassland Soils"

_microorganisms, 2021, doi:10.3390/microorganisms9081677_

Round 1

Reviewer 1 Report

The article entitled "Drought and plant community composition affect the metabolic and genotypic diversity of Pseudomonas strains in grassland soils" is very interesting and the objectives of the study are novel. The authors have executed a well-organized and executed experiment in order to test all the questions posed in the introduction. The lab analyses methods as well as the statistical analyses applied to their dataset were well performed. the discussion is also very well constructed. I do suggest the publication of this article as soon as the authors address the following issues.

  1. the authors must provide in the M&M section some information regarding the soil characteristics. since certain soil parameters affect significantly the microbial community it is very important to clarify that no significant differences existed among sites.
  2. the description of the experimental design is quite complicated. There are 80 plots yet the in the study only 30 were used. So why do they have to mention all of them. Also the Table S1 has this issue too as it is difficult for the reader to identify the plots that were used.
  3. In supplementary materials, a figure presenting the moisture percentages with standard errors must be included.
  4. In all text and supplementary materials, all scientific names must be in italics.

Author Response

Reviewer 1:

Comments and Suggestions for Authors

The article entitled "Drought and plant community composition affect the metabolic and genotypic diversity of Pseudomonas strains in grassland soils" is very interesting and the objectives of the study are novel. The authors have executed a well-organized and executed experiment in order to test all the questions posed in the introduction. The lab analyses methods as well as the statistical analyses applied to their dataset were well performed. the discussion is also very well constructed. I do suggest the publication of this article as soon as the authors address the following issues.

We highly appreciate the positive feedback for our study and additionally thank the reviewer for the useful hints to further improve the quality of the manuscript.

  1. the authors must provide in the M&M section some information regarding the soil characteristics. since certain soil parameters affect significantly the microbial community it is very important to clarify that no significant differences existed among sites.

We thank the reviewer for the valuable comment and included both, information on the soil (Chernozem) used for the experiment (section 2.1) as well as information on treatments and season-dependent soil properties (section 2.2, Table S3).

  1. the description of the experimental design is quite complicated. There are 80 plots yet the in the study only 30 were used. So why do they have to mention all of them. Also the Table S1 has this issue too as it is difficult for the reader to identify the plots that were used.

We thank the reviewer for the valuable comment and focus much more on the sampled plots. Nevertheless, it is necessary to show and mention the full design of the used experiment, to understand the experimental setup and the scheme of the experiment in Figure S1. For a better identification we provided the Figure S1 and marked the plots used within the study (red asterisk in Figure S1). We agree with the reviewer, that the information provided in Table S1 might be too much and confusing. We removed the plots not used within the study in the Table S1.

  1. In supplementary materials, a figure presenting the moisture percentages with standard errors must be included.

The reviewer is right and we thus we added, in addition to the information provided by Table S3, a new figure (Fig. S2), showing the soil moisture contents (Season x Drought Treatment).

  1. In all text and supplementary materials, all scientific names must be in italics.

We carefully checked the manuscript for scientific species names and changed the style to italics.

Reviewer 2 Report

The climate change effects on natural microbial populations is a subject of great interest. I read the manuscript with interest and found that many results are merely variations of data published more than ten years ago (in particular those concerning the influence of plant species on microbial populations).

There are some specific issues which I would like the authors give some explanations:

1) I am confused about the isolation procedure of bacterial strains. Authors stated that they isolated PGPR pseudomonads. Plant growth promoting rhizobacteria are generally isolated from plant roots and subsequent specific tests define their plant growth promoting ability. In this work bacteria were isolated from soil samples and were not subjected to specific plant growth promotion tests. I think authors should use a more sober, precise and less confusing language when addressing the characteristics of these bacterial strains. 

2) There is a lot of statistics and whenever a significant difference is found, automatically a biological significance is attached to it. In fig. 3, I do not see any great differences, except just in one case. So, authors should interpret more deeply significant statistical deifferences and discuss whether they have also a real biological counterpart.

3) The Discussion section is too long, should be more concise.

4) Last but not least, a very important data is lacking, necessary for the overall evaluation of the role of pseudomonads: quantification of total bacterial population and of pseudomonas population.

Round 2

Reviewer 2 Report

The manuscript has been sufficiently improved to warrant publication